# Complete Genome Sequence, Genome Stability and Phylogeny of the Vaccine Strain *Mycobacterium bovis* BCG SL222 Sofia

**DOI:** 10.3390/vaccines9030237

**Published:** 2021-03-09

**Authors:** Stefan Panaiotov, Yordan Hodzhev, Vladimir Tolchkov, Borislava Tsafarova, Alexander Mihailov, Tzvetelina Stefanova

**Affiliations:** 1National Center of Infectious and Parasitic Diseases, 1504 Sofia, Bulgaria; y.hodzhev@ncipd.org (Y.H.); tolchkov@ncipd.org (V.T.); b.tsafarova@ncipd.org (B.T.); 2BB-NCIPD Ltd., 1504 Sofia, Bulgaria; bcg@bulbio.com (A.M.); t.stefanova@bulbio.com (T.S.)

**Keywords:** BCG vaccine, BCG sub-strains, BCG-I, BCG SL222 Sofia, complete genome, phylogeny

## Abstract

*Mycobacterium bovis* bacillus Calmette–Guérin (BCG) is the only live attenuated vaccine available against tuberculosis. The first BCG vaccination was done exactly 100 years ago, in 1921. The BCG vaccine strains used worldwide represent a family of daughter sub-strains with distinct genotypic characteristics. BCG SL222 Sofia is a seed lot sub-strain descending from the Russian BCG-I (seed lot 374a) strain and has been used for vaccine production in Bulgaria since 1972. Here, we report the assembled circular genome sequence of *Mycobacterium bovis* BCG SL222 Sofia and phylogeny analysis with the most closely related BCG sub-strains. The full circular genome of BCG SL222 Sofia had a length of 4,370,706 bp with an average GC content of 65.60%. After 49 years of in vitro evolution in a freeze-dried condition, we identified four SNP mutations as compared to the reference BCG-I (Russia-368) sequence. BCG vaccination is of central importance for the TB elimination programs in many countries. Since 1991, almost 40 million vaccine doses of the BCG SL222 Sofia have been distributed annually through the United Nations Children’s Fund (UNICEF) and the Pan American Health Organization (PAHO) to approximately 120 countries. The availability of the complete reference genome sequence for *M. bovis* BCG SL222 Sofia, a WHO reference reagent for the Russian BCG-I sub-strain, will facilitate the identity assurance of the genomic stability, will contribute to more consistent manufacturing, and has an important value in standardization and differentiation of sub-strains used in vaccine production. We propose to rename the sub-strain BCG SL222 Sofia to BCG-Sofia for practical and common use.

## 1. Introduction

The BCG vaccine sub-strains currently in use are all progenies of the original strain attenuated by Calmette and Guérin during 1908–1921 [1]. In the absence of lyophilization or freezing and the production of seed-lots until the 1960s, the propagation of BCG through continuous passage under different laboratory conditions resulted in the generation of daughter BCG sub-strains with different biochemical, morphological, and immunological characteristics. Among the family of BCG sub-strains unique molecular profiles were detected [2,3]. This fact suggests ongoing in vitro microevolution of the BCG sub-strains that started in 1908. After a century of mass use, BCG vaccine demonstrated a duration of protection of at least ten years with residual vaccine effectiveness up to 20–25 years [4].

Today, five WHO reference BCG vaccine preparations, namely Danish (Copenhagen—1331), Japanese (Tokyo—172-1), Russian (Moscow–368), Brazilian (Moreau/RDJ), and Bulgarian (SL222 Sofia), are in mass production and use [5,6]. Examination of historical records showed that the BCG SL222 Sofia sub-strain is derived from generation 374a of BCG-I [7,8]. The production of lyophilized stocks started in Russia in the 1940s, and the seed lot system was introduced in 1954 [9]. The BCG-I (seed lot 374a) lyophilized batch was produced in February 1971 in Moscow, and samples of it were sent to Bulgaria. The strain was received at the National Center of Infectious and Parasitic Diseases for the production of freeze-dried live BCG vaccine. Primary seed lot (Master Seed) SL222 Sofia of the Bulgarian BCG vaccine was produced in May 1972 directly from generation 374a of the Russian strain BCG-I. The Russian strain itself was originally received from the Pasteur Institute in Paris in 1925 by Dr. Tarassevitch at the State Institute of Standardization and Control of Medical and Biological Products, Moscow, USSR, and designated BCG-I [10,11]. According to these historical data we classified BCG SL222 Sofia as belonging to the ‘early’ strains such as BCG-I (Moscow-368), BCG-Japan (Tokyo–172-1), and BCG-Moreau RDJ distributed in 1924–1925. These strains maintain two copies of the IS6110 and mpt64 gene. The strains distributed in 1925–1926 BCG-Sweden and BCG-Birkhaug lost one copy of IS6110. The third most abundant group of BCG sub-strains, so-called ‘late strains’ distributed between 1926 and 1931, includes BCG-Denmark, BCG-Phipps, BCG-Tice, BCG-Flappier, and BCG-Pasteur, which have lost the mpt64 gene and have one copy of IS6110 [2].

BCG is probably the most widely applied vaccine in modern medicine. There is no doubt that BCG will continue to play a major role in TB control. It remains part of upcoming clinical trials as an integral component of new vaccines or as a primer to be boosted by new components [12]. In the last few years the importance of BCG characterization and control has been strongly emphasized, especially in the light of new knowledge based on the development of whole-genome sequencing and analysis techniques [13]. BCG SL222 Sofia is of central importance to the vaccination programs in Bulgaria and many other countries around the world. Since 1991 the BCG SL222 Sofia vaccine has been approved by WHO for the UN supply. In 2009 BCG SL222 Sofia was established as the 1st WHO Reference Reagent for BCG vaccine of Russian BCG-I sub-strain [14,15,16]. The rationale for complete genome sequencing of BCG SL222 Sofia aims to prove genetic stability of the strain after decades of storage, to provide scientific evidence with respect to BCG strains’ phylogeny, and to contribute to the global consultations on the WHO requirements for production and control of BCG vaccines.

## 2. Materials and Methods

The study was planned in agreement with the implemented system for quality control by BB-NCIPD Ltd., which meets the requirements of ISO 9001:2000 standard (Certificate Lloyd’s Register QA No. 368090). BB-NCIPD Ltd. holds production license No. I-65/12.02.2003, issued by the Bulgarian Drug Agency, which approves it as a manufacturer, who meets the requirements of Human Medicines and Pharmacies Act.

### 2.1. Origin of the BCG SL222 Sofia Sub-Strain

The Master seed lot BCG SL222 Sofia was manufactured according to the requirements for dried BCG vaccine (WHO Technical Report Series) at the Laboratory for BCG Vaccine, National Centre of Infectious and Parasitic Diseases; 26, Y. Sakazov Blvd., 1504 Sofia, Bulgaria. Microbial growth was performed on Glycerinated potato medium [17] (BB-NCIPD Ltd., Sofia, Bulgaria) and Sauton medium [18] (BB-NCIPD Ltd., Bulgaria). Four passages were performed from the primary seed BCG-I seed lot 374a. The temperature of incubation was 37 °C. The harvest was performed using nine days’ bacterial surface culture on Sauton medium. The date of manufacture of BCG SL222 Sofia was 13 May 1972. According to the requirements described in WHO Technical Report Series, the Master seed lot BCG SL222 Sofia was verified as *Mycobacterium bovis* BCG by the morphological appearance of the bacilli in stained smears (Ziehl–Neelsen) and by the characteristic appearance of the colonies grown on Löwenstein–Jensen medium (BB-NCIPD Ltd., Bulgaria). The Master seed lot BCG SL222 Sofia was lyophilized and stored at −20 °C until 2008 when WHO recommended storage of the BCG Master seed lots at −80 °C.

### 2.2. DNA Isolation and Complete Genome Sequencing Approach

For DNA isolation we used a fresh culture from 49-year-old freeze-dried master seed lot BCG SL222 Sofia resuspended in saline and grown on Löwenstein–Jensen medium for 35 days at 37 °C. The DNA was isolated according to the procedures described by Ausubel [19] and van Soolingen [20] with the following modifications. In short, three full loops of fresh microbial mass were collected and placed in 400 µL TE buffer (20 mM Tris-HCl, 1 mM EDTA (pH 8.0)) in a 1.5 mL tube. The cell suspension was vortexed and heated at 80 °C for 20 min to kill the cells. Lysozyme, to a final concentration of 1 mg/mL, and 1 U of RNase H (Thermo Fisher Scientific, Waltham, MA, USA) were added, and the tube was incubated for 24 h at 37 °C. Seventy microliters of 10% sodium dodecyl sulfate (SDS) (Sigma-Aldrich, St. Louis, MO, USA) and 6 µL of proteinase K (10 mg/mL concentration) (Sigma-Aldrich, USA) were added, and the mixture was incubated for 1 h at 65 °C. One hundred microliters of 5 M NaCl and 80 µL of prewarmed at 65 °C CTAB/NaCl (10% N-cetyl-N,N,N-trimethylammonium bromide/0.73 M NaCl) solutions were added. The suspension was homogenized by mixing by hand and incubated for 30 min at 65 °C. An equal volume of phenol-chloroform-isoamyl alcohol (25:24:1, vol/vol) was added, and the cell lysate was mixed by hand to keep the isolated DNA as intact as possible. After centrifugation for 5 min at 12,000× *g*, the supernatant was transferred to a new 1.5 mL tube. A second extraction with chloroform-isoamyl alcohol (24:1, vol/vol) was performed and the sample centrifuged. The supernatant was transferred to a new tube, and 0.7 volume of isopropanol was added to precipitate the DNA. After 24 h at −20 °C and centrifugation for 15 min at 12,000× *g*, the DNA pellet was washed once with 70% ethanol, again centrifuged, the air-dried pellet was re-dissolved in 500 µL of sterile DNA/RNA free water by gently tapping the tube, and it was left at 4 °C for 24 h before spectrophotometric quantitation. The DNA quality was 1.86 OD measured at 260/280 nm (UV/Vis spectrophotometer QeneQuant 1300, GE Healthcare, Chicago, IL, USA).

### 2.3. Genome Sequencing and Assembly

To sequence and assemble the complete genome sequence of BCG SL222 Sofia we combined the second-generation for short-read sequencing (Illumina, San Diego, CA, USA) and third-generation for long-read sequencing (Nanopore) approaches and bioinformatics analysis. The isolated DNA was used for both Illumina and Nanopore library preparation. An Illumina sequencing library was prepared using the Nextera DNA Flex kit and Nextera DNA CD indexes (Illumina) according to the manufacturer’s protocol. Sequencing was performed in 2 × 300 bp cycles on an Illumina MiSeq sequencer using the MiSeq Reagent Kit v3. Quality control and data processing for filtering, formatting and trimming were performed by the use of prinseq-lite program (v.0.20.4, http://prinseq.sourceforge.net (accessed on 9 March 2021)) [21] applying the following parameters: min_length, 50; trim_qual_right, 30; trim_qual_type, mean; trim_qual_window, 20. The total number of reads was 15,084,680. Reads passing quality check (96.49% of total samples reads) or the total number of cleaned reads were 14,555,656. The total number of joined reads was 13,451,084 (92.41% of cleaned reads). Sequences from Illumina sequencing were joined using the FLASH program applying default parameters [22]. Illumina reads were loaded into Geneious Prime software version 2020.2.4 [23]. Reads were trimmed with the BBDuk plugin (v.1.0, https://sourceforge.net/projects/bbmap (accessed on 9 March 2021)). Adapters on the right ends and low-quality ends (quality below 20%) were trimmed, while reads shorter than 200 bp were discarded. Then the reads were subjected to preprocessing using the Geneious software package. As a reference genome BCG-I (Russia-368) NZ_CP013741 was used, and BCG SL222 Sofia genome assembly was performed by using the Geneious “Map to reference” tool.

Tandem repeats and high GC content regions in BCG led to low coverage after genome mapping with Illumina technology. The second run of sequencing to generate long reads for scaffolding was performed based on Nanopore PromethION technology (Oxford, UK). The Nanopore data processing software Guppy (v. 3.03) was applied for base calling. The online NanoPlot (v.1.30.1) service (http://nanoplot.bioinf.be (accessed on 9 March 2021)) was applied for quality controls. The QC showed that the total bases were 123,155,602; the number of reads was 139,414; the median read length was 604; the read length N50 was 1178; the median read quality was 11; the number and percentage of reads quality over than 7 were 133,439 (95.7%). Omitting the preprocessing steps, Nanopore reads were subjected to reference mapping with Geneious Prime bioinformatics software sequence analysis tools.

### 2.4. Phylogeny Analysis

The phylogenetic tree was calculated to represent the evolutionary history of BCG sub-strains using the RAxML (Randomized Axelerated Maximum Likelihood) algorithm built within the Geneious prime platform [24]. Ten completely sequenced genomes of BCG sub-strains were employed with *M. bovis* (LT708304) genome as an outgroup (Table 1). The rate of nucleotide substitutions per genome was calculated. Data are schematically represented in Figure 2.

## 3. Results

Complete genome sequencing of mycobacteria and BCG sub-strains is challenged by the occurrence of large genome segments of a variable number of tandem repeats (VNTR) and a high GC content (65%). To overcome these technical obstacles we combined second- and third-generation sequencing approaches and bioinformatics to construct the completely assembled genome sequence of BCG SL222 Sofia. Using two sequencing approaches gave higher certainty with low-level variant calls, minimizing the need for subjective correction of repetitive regions, mobile elements, sequence gaps, or invalid sequences.

Left and right ends of the assembled sequence overlapped, and the circular molecule of the BCG SL222 Sofia was confirmed visually. The complete genome sequence of BCG SL222 Sofia comprised one circular chromosome consisting of 4,370,706 bp with an average GC content of 65.60%. Prediction of bacterial protein-coding genes, structural RNAs, tRNAs, and other functional genome units was generated with the NCBI Prokaryotic Genome Annotation Pipeline (PGAP, https://github.com/ncbi/pgap (accessed on 9 March 2021)). A total of 4078 protein-coding genes, 3 rRNAs, and 45 tRNAs were predicted and identified in the BCG SL222 Sofia genome.

The genome sequence of BCG SL222 Sofia is one base longer and revealed 4 single nucleotide polymorphisms (SNPs) as compared to BCG-I (Russia-368) NZ_CP013741 as a reference strain. The sequence of the parent BCG seed lot 374a strain is not available. It could be supposed that SNP mutations might have occurred among different seed lots of BCG-I [9]. Our sequencing data could be considered closest to the original sequence of BCG-I seed lot 374a. BCG SL222 Sofia is the fourth passage of BCG-I seed lot 374a.

At present BCG vaccine production in Russia is based on seed lot 368 produced in 2006. Freeze-dried seed lot 368 derived from seed lot 359 lyophilized in 1963 [9]. We detected four microevolution events in BCG SL222 Sofia as compared to BCG-I seed lot 368 (Table 2).

We identified one G deletion at position 841,951 (BCG-I reference genome sequence NZ_CP013741 within repeat (G)6 leading to (G)5 (coverage 137, variant frequency 98.5%)), which resulted in a frameshift in encoded PE-family protein. Two SNP insertions in short single nucleotide repeats were also found. The first was a C insertion at genome position 2,397,101 in repeat (C)4 leading to (C)5, causing frameshift in encoded PE-family protein (coverage 137, variant frequency 76.06%). The second was a G insertion at position 3,912,049 within a (G)4 repeat leading to (G)5 and resulted in a frameshift in protein-PII uridylyltransferase (coverage 71, variant frequency 91.5%). Finally, there was a transition A -> G at position 3,175,530, which led to synonymous codon change GCA -> GCG in a PE family protein (coverage 97, variant frequency, 96.9%) (Table 2).

All characteristic sequence features of BCG-I were confirmed for BCG SL222 Sofia. RD1 (Region of Difference) was deleted in all BCG sub-strains including BCG SL222 Sofia. The unique deletion BCG-I RDRussia of 1603 bp (Rv3697c to Rv3698) according to the location in the *M. tuberculosis* H37Rv genome [3,25] was also present in BCG SL222 Sofia [26]. The RDRussia deletion was 1608 bp as compared to the reference *M. bovis* genome sequence LT708304. The function of the deleted region is not fully known, but the encoded product is a membrane protein, possibly linked to the structure of the cell wall. RD1 and RDRussia form deletions of more than 11 kbp affecting 11 ORFs in the reference sequence of virulent *M. bovis* (LT708304). Both deletions are located at the end of the genome. Except for BCG Tokyo-172, BCG Sweden, and BCG Birkhaug, which have only RD1 deletion affecting 9 ORFs, all other BCG sub-strains have longer deletions from 10 to 20 kbp as compared to BCG SL222 Sofia. According to these general calculations, BCG SL222 Sofia sub-strain might be more attenuated, less virulent, and immunogenic as compared to BCG Tokyo-172, but more virulent and/or immunogenic as compared to BCG Pasteur-1173P or BCG Danish-1331 where affected ORFs are 31 and 22, respectively, and over-attenuation in these sub-strains could be supposed. There are no detailed comparative studies. A characteristic feature of the ‘early’ BCG sub-strains, including BCG SL222 Sofia, is the presence of RD2. DU2 duplication with four copies of marker gene sigH and other repeated regions (phoP) was found in the genome of the Bulgarian BCG sub-strain. DU2 is the same duplication found in the BCG-I genome in previous observations and is related to the catabolic activity of some enzymes [8,25,27]. We confirm the presence of the full sequence of RD14 of 9073 bp (including leuA repeat region of 1517 bp), which in some BCG sub-strains, like BCG-Pasteur is absent. The mobile elements IS6110 and IS1547 are present in two copies and are a characteristic feature of BCG-I and BCG SL222 Sofia. Both sub-strains BCG-I and BCG SL222 Sofia feature 525 bp deletion of the polyketide synthase 12 (pks12) gene, necessary for β-phosphomycoketide production and the CD1c-mediated T cell response [28]. Pyrazinamide drug resistance is a characteristic feature for all *Mycobacterium bovis* strains including BCG. Pyrazinamide resistance in BCG SL222 Sofia was identified in the pncA gene as a single substitution C169G leading to His57Asp mutation.

Full genome sequencing confirmed that the genome stability of BCG SL222 Sofia was not affected by the lyophilization procedure, nor by the storage conditions of the freeze-dried ampoules for more than 45 years at −20 °C until 2008 and at −80 °C since then.

The annotated genome sequence of BCG SL222 Sofia was deposited under GenBank accession number CP064405. The version described is the first version, CP064405.1. Illumina raw reads accession number is SRX9465445 and for Nanopore is SRX9523320. The corresponding bioproject number is PRJNA673391.

## 4. Discussion

The BCG sub-strains are considered to have evolved mainly by the accumulation of gene deletions (RDs) and gene amplifications (DU1 and DU2). The main deletions bigger than 1000 bp and insertions bigger than 800 bp leading to attenuation in BCG SL222 Sofia as compared to the reference *M. bovis* sequence are presented in Table 3.

It was found that the genome of *M. bovis* BCG SL222 Sofia underwent the same genetic events as its closest predecessor BCG-I seed lot 368, with only four SNP genetic variations during subculturing and storage in Bulgaria. These minimal genetic variations do not affect essential phenotypic characteristics of BCG SL222 Sofia. The presence of an RD2 region in its genome places BCG Sofia in the group of BCG sub-strains closest to the original Calmette and Guérin strain. It has been shown that the region called RDRussia, which covers 1608 bp, absent in the BCG-I genome, is also missing from BCG Sofia.

Historically and evolutionary BCG SL222 Sofia is most closely related to BCG-I (Moscow-368) and BCG-Japan (Tokyo—172-1). Examination of historical records showed that the Birkhaug, Brazil, Japan, Sweden, and Russia BCG sub-strains emerged soon after the distribution of cultures derived from the 1921 parent strain developed by Calmette and Guérin. The BCG SL222 Sofia strain emerged in 1972 from the Russian BCG-I strain seed lot 374a produced in 1971 in Moscow. Thus, BCG SL222 Sofia is a daughter strain derived from BCG-I Russia (Figure 1).

BCG SL222 Sofia contains an RD2 region, which is typical for all ‘early’ BCG strains. All BCG strains have lost the RD1 region, which has been associated with attenuation from virulence. These ‘early’ strains are also characterized by duplication DU2, a large fragment containing essential genes, and absence of DU1. BCG-Denmark, BCG-Tice, and BCG-Glaxo sub-strains that were distributed after 1931 have only one copy of IS6110 and have lost RD2. These BCG sub-strains underwent ‘late’ deletions suspected to be responsible for BCG’s over-attenuation. The BCG phylogeny based on the nucleotide substitution rate among the BCG sub-strains with known complete genome sequence is presented in Figure 2. The reported data in Figure 2 are supported by historical [1], deletions mapping [27,28], and sequencing data [25].

The presence of an SNP insertional mutation at position 413 bp from the start of the recA gene was previously identified by Keller et al., (2008), describing BCG-I (Russia) as a natural recA mutant [29]. We did not confirm for BCG SL222 Sofia insertional mutation C414 at the recA gene. Mutation C414 is not present in BCG-I as confirmed by Narvskaya et al. [9]. The ‘early’ BCG sub-strains including BCG SL222 Sofia sequence showed retention of mpt64 and two copies of IS6110.

Subculturing of BCG sub-strains may lead to the accumulation of genetic changes and over-attenuation of the sub-strains. WHO recommended in 1966 that the BCG vaccine should not be prepared from any culture that had gone more than 12 passages starting from a defined freeze-dried seed lot [30]. The Bulgarian BCG SL222 Sofia sub-strain (Master seed) is produced from the fourth passage after the Russian BCG-I strain seed lot 374a, followed by three passages between master seed lot and working seed lot and another two passages between working seed and final lot. That means the Bulgarian BCG vaccine is produced from the ninth passage, reckoned of the original source.

BCG immunization of newborns and infants is obligatory in many countries. In Bulgaria, BCG vaccine is given to all healthy newborns, not earlier than 48 h after the birth. Children between 7 and 10 months of age without BCG scar are subject to BCG immunization after a negative tuberculin test. BCG reimmunization is given to all tuberculin-negative children at 7 years of age. It is well known that in addition to its specific anti-tuberculosis effect, the BCG vaccine has some heterologous protective effects. It has been proven that the BCG vaccine may give non-specific protection to other microbial or viral infections [31], and even cancer protection [32]. It was recently confirmed that in Bulgaria, children and adolescents of age group 0–19 years with COVID-19 are 3.5% of all confirmed cases (as at 10 February 2021). Three of them were fatal (0.03%). All other age groups demonstrate a higher incidence of COVID-19 and fatal cases. Future research is needed to confirm any positive immunomodulation effects of BCG vaccine against pathogens other than *M. tuberculosis* and *M. bovis.*

## 5. Conclusions

BCG vaccination will continue to play a major role in TB control worldwide. Our approach using massive parallel sequencing by applying second- and third-generation technologies provides both high per-bp accuracy, allowing for SNP calling, and complete resolution of the genome assembly across large repeat regions common in BCG.

Identified four SNPs between BCG SL222 Sofia and BCG-I seed lot 368 do not affect essential genes leading to phenotypic specificity. Our results confirm that BCG SL222 Sofia (1972) is genetically identical to BCG-I Russia seed lot 368 (2006).

The availability of the complete reference genome sequence for *M. bovis* BCG Sofia, a WHO reference reagent for BCG-I Russia, will facilitate the identity assurance of the genomic integrity, will contribute to more consistent manufacturing, and has an important value in standardization and differentiation of BCG sub-strains used in vaccine production.

As seed lot 222 has been in use for mass production of the BCG vaccine since 1972, we propose to denote the sub-strain BCG SL222 Sofia as BCG Sofia, a shorter name for routine practical use.

## Figures and Tables

**Figure 1 vaccines-09-00237-f001:**
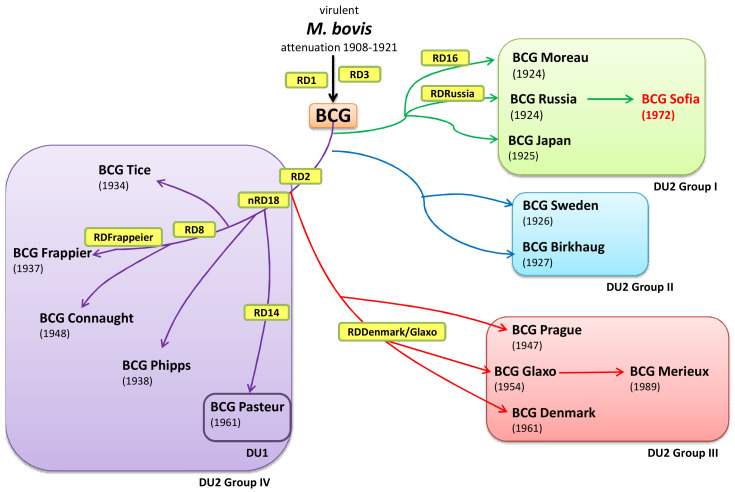
Evolution of the BCG sub-strains according to historical and genetic data. Phylogenetic positioning of BCG SL222 Sofia among the BCG sub-strains.

**Figure 2 vaccines-09-00237-f002:**
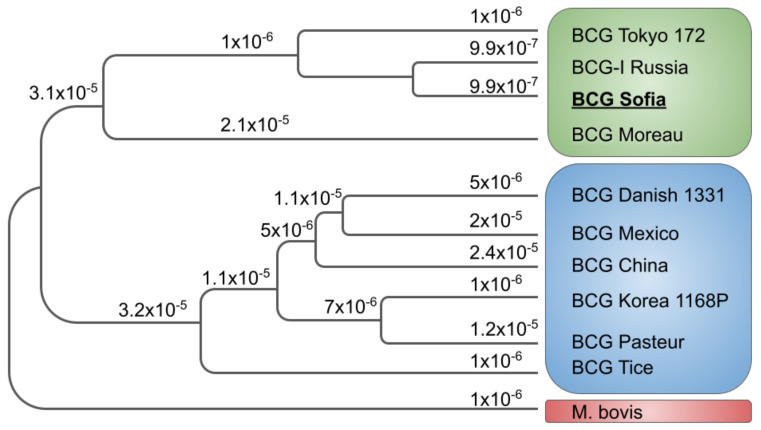
Randomized Axelerated Maximum Likelihood (RAxML) phylogenetic tree of 10 completed BCG genomes. The topology is rooted with *M. bovis* (LT708304). Over each branch is annotated the calculated nucleotide substitution rate. The “early” and the “late” groups of BCG are represented in green and blue, respectively.

**Table 1 vaccines-09-00237-t001:** Name, GenBank reference number and size of used BCG sub-strains with complete genome sequence for phylogenetic analysis.

BCG Sub-Strain	GenBank Accession Number	Length (bp)
BCG-I Russia-368	NZ_CP013741	4,370,705
BCG SL222 Sofia	CP064405	4,370,706
BCG Danish 1331	NZ_CP039850	4,411,814
BCG Tice	NZ_CUWQ01000001	4,317,625
BCG China	NZ_CUWG01000001	4,318,346
BCG Tokyo 172	CP014566	4,371,707
BCG Moreau RDJ	NZ_AM412059	4,339,996
BCG Mexico	CP002095	4,350,386
BCG Pasteur 1173P2	NC008769	4,374,550
BCG Korea 1168P	NC020245	4,376,727
*Mycobacterium bovis AF212297*	LT708304	4,349,904

**Table 2 vaccines-09-00237-t002:** Single nucleotide polymorphisms (SNPs) found in BCG SL222 Sofia when using BCG-I assembly NZ_CP013741 as a reference. Coordinates and frameshifted amino acids were derived by automated computational analysis by SNP calling prediction method built in Geneious Prime software.

Position (bp)	BCG-I SL368	BCG SL222 Sofia	Type of Mutation	Ammino Acid Change	Gene
841,951	(G)6	(G)5	Non-synonymous deletion	Frameshift Gly -> Arg	PE family protein
2,397,101	(C)4	(C)5	Synonymous insertion	Frameshift Gly -> Gly	PE family protein
3,175,530	A	G	Synonymous transition	Ala -> Ala	Protein-PII uridylyltransferase
3,912,049	(G)4	(G)5	Synonymous insertion	Frameshift Gly -> Gly	PE family protein

**Table 3 vaccines-09-00237-t003:** Comparative genomics analysis of BCG-Sofia as compared to reference *M. bovis* genome sequence LT708304.

DELETIONS > 1000 bp			
Nucleotide Position	Size (bp)	Affected CDS	Genes/Proteins
1,415,767–1,420,151	4384	3 ORFs	*tbd2*, *pknH1*, *pknH2*
1,772,330–1,781,585	9255	14 ORFs	RD3, probable phage proteins
2,612,546–2,615,131	2585	2 ORFs	*ppe40*, *ppe71*
4,130,922–4,132,530	1608	3 ORFs	RDRussia—glutamate-cysteine ligase
4,340,448–4,349,958	9510	9 ORFs	RD1—*esx-1* (secretion associated protein), *esat-6*, *espk*, *cfp10*
**INSERTIONS > 800 bp**			
1,761,134–1,761,946	812	2 ORFs	Putative acyltransferase
3,669,049–3,710,643	41,594	42 ORFs	DU2 region
4,361,253–4,363,666	2413	3 ORFs	Type VII secretion protein EccB, insertion not found in BCG-Moreau

## Data Availability

The complete genome sequence of BCG SL222 Sofia is deposited at NCBI GenBank. Accession numbers of other BCG substrains complete genome sequences are reported throughout the text with corresponding hyperlink.

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
