# Peer review of "Complete Genome Sequence, Genome Stability and Phylogeny of the Vaccine Strain Mycobacterium bovis BCG SL222 Sofia"

_vaccines, 2021, doi:10.3390/vaccines9030237_

Round 1

Reviewer 1 Report

Dear Authors

In this study, authors report the assembled circular genome sequence of Mycobacterium bovis BCG SL222 Sofia and phylogeny analysis with the most closely related BCG sub-strains.

The paper is well written and interesting to read. The manuscript is within the journal's scope.

The Introduction is concise. The purpose of the study is clearly defined. The interpretation of the results is correct. The figures and tables have clear structure. The discussion interpretations and conclusions justified by the results of the study.

Minor revisions

Abstract Line 19 Replace   “Mycobacteium“ With “Mycobacterium

Introduction Line 53 Replace   “in-vitro“ With “in vitro

Material and methods – 2.1 and 2.2 The used equipment’s should be clearly stated including Brand names, models and country.

Author Response

Dear Reviewer,

Thank you for your review of our manuscript.

Point 1: Abstract Line 19 Replace   “Mycobacteium“ With “Mycobacterium

Response 1: Introduced correction.

Point 2: Introduction Line 53 Replace   “in-vitro“ With “in vitro

Response 2: Introduced correction.

Point 3: Material and methods – 2.1 and 2.2 The used equipment’s should be clearly stated including Brand names, models and country.

Response 3: Appropriate corrections were inroduced to section ‘Materials and Methods’ – 2.1 and 2.2.

Reviewer 2 Report

The manuscript reports the genome sequence of Mycobacteium bovis BCG SL222 vaccine sub-strain and assures its stability over 49 years of in vitro evolution in a freeze-dry condition, with 4 single nucleotide polymorphism mutations identified as compared to the reference sequence. 

A few minor comments to address: 

This introduction into the history of BCG development, while quite interesting, seems to be redundant for the research article, not a review. On Page 1, lines from 36 to 48 and 56 to 75 should be removed and starting from the appearance of daughter BCG sub-strains and the current vaccine preparations available. Overall, the manuscript length is excessive and requires some shortening revisions. 

On Page 9, lines 353-355 the authors speculate about BCG vaccination to attenuate COVID-19 ad by some reasons, refer to some totally irrelevant study back from 2008 (#29, Keller et al., BMC Microbiol 2008). Indeed, there were a few recent works claiming the effect of BCG on SARS-CoV-2 seroprevalence (Escobar et al., PNAS, 2020; Rivas et al., J Clin Invest, 2021). However, most of the data were not confirmed and criticized by specialists (Lindestam Arlehamn et al., PNAS, 2020). I would suggest removing all the unnecessary speculations on BCG relations to COVD-19 epidemiology or physiology. 

Author Response

Dear Reviewer,

Thank you for your review of our manuscript.

Point 1: This introduction into the history of BCG development, while quite interesting, seems to be redundant for the research article, not a review. On Page 1, lines from 36 to 48 and 56 to 75 should be removed and starting from the appearance of daughter BCG sub-strains and the current vaccine preparations available. Overall, the manuscript length is excessive and requires some shortening revisions. 

Response 1: On Page 1, lines from 36 to 48 and 56 to 75 were removed. Ref. 5 and 6 were moved and adapted to: Examination of historical records showed that the BCG SL222 Sofia sub-strain is derived from generation 374а of BCG-I [7,8].

Point 2: On Page 9, lines 353-355 the authors speculate about BCG vaccination to attenuate COVID-19 ad by some reasons, refer to some totally irrelevant study back from 2008 (#29, Keller et al., BMC Microbiol 2008). Indeed, there were a few recent works claiming the effect of BCG on SARS-CoV-2 seroprevalence (Escobar et al., PNAS, 2020; Rivas et al., J Clin Invest, 2021). However, most of the data were not confirmed and criticized by specialists (Lindestam Arlehamn et al., PNAS, 2020). I would suggest removing all the unnecessary speculations on BCG relations to COVD-19 epidemiology or physiology. 

Response 2: Ref. 29 is a wrong citation remained from previous revision. As suggested we removed all the unnecessary speculations on BCG relations to COVD-19 epidemiology or physiology. We removed the sentence: A recent study demonstrated that BCG immunization coverage, especially for the most recent 15 years, contributed to the attenuation of the spread and severity of the COVID-19 pandemic [29].

Ref. 32 and 34 were also removed.

Reviewer 3 Report

vaccines-1141355

Complete Genome Sequence, Genome Stability and Phylogeny of the Vaccine Strain Mycobacterium bovis BCG SL222 Sofia

Stefan Panaiotov , Yordan Hodzhev , Vladimir Tolchkov , Borislava Tsafarova , Alexander Mihailov , Tzvetelina Stefanova

This is a very good study and well written manuscript. This paper is ready for publication in vaccines. There were three grammatical or spelling issues and one method issue save listed below.

Methods: Please include institutional compliance or approval.

Page 2, line 45: Change sequel to sequelae.

Page 4, line 166: Change Reeds to Reads.

Page 6, Lines 255 – 258: This is an awkward sentence. I think a comma after ORFs will make it clearer.

Author Response

Dear Reviewer,

Thank you for your review of our manuscript.

Point 1: Methods: Please include institutional compliance or approval.

Response 1: The following clarification was introduced in Materials and Methods section - The study was planned in agreement with the implemented system for quality control by BB-NCIPD Ltd., which meets the requirements of ISO 9001:2000 standard (Certificate Lloyd's Register QA No. 368090). BB-NCIPD Ltd. holds production license No. I-65/12.02.2003, issued by the Bulgarian Drug Agency, which approves it as a manufacturer, who meets the requirements of Human Medicines and Pharmacies Act.

Point 2: Page 2, line 45: Change sequel to sequelae.

Response 2: The paragraph was deleted as suggested by another reviewer who considers that  the introduction into the history of BCG development seems to be redundant for our research article.

Point 3: Page 4, line 166: Change Reeds to Reads.

Response 3: Introduced correction.

Point 4: Page 6, Lines 255 – 258: This is an awkward sentence. I think a comma after ORFs will make it clearer.

Response 4: Introduced correction.

Reviewer 4 Report

In general, the manuscript was well-written, easy to follow, and scientifically sound. There were some sentences with grammar issues.  Lines 355 - 357 need to rewritten.  The authors also need to check the references. Line 355 refers to reference #29, but it should probably be #34. Personally, I think the discussion of protection to the SARS-CoV-2 virus is not needed, and is tangential to the entire discussion.  

Author Response

Dear Reviewer,

Thank you for your review of our manuscript.

Point 1: Lines 355 - 357 need to rewritten. 

Response 1: The sentence was rewritten: It was recently confirmed, that in Bulgaria, children and adolescents of age group 0-19 years with COVID-19 are 3.5% (7949 cases as at Feb. 2021). Three of them were fatal (0.03%). (0.03%).

Point 2: The authors also need to check the references. Line 355 refers to reference #29, but it should probably be #34. Personally, I think the discussion of protection to the SARS-CoV-2 virus is not needed, and is tangential to the entire discussion. 

Response 1: Ref. 29 is a wrong citation. Another reviewer also suggested to remove all the unnecessary speculations on BCG relations to COVD-19 epidemiology or physiology. We removed the sentence: A recent study demonstrated that BCG immunization coverage, especially for the most recent 15 years, contributed to the attenuation of the spread and severity of the COVID-19 pandemic [29].

Ref. 32 and 34 were also removed.